# Internet Gaming Disorder Clustering Based on Personality Traits in Adolescents, and Its Relation with Comorbid Psychological Symptoms

**DOI:** 10.3390/ijerph17051516

**Published:** 2020-02-26

**Authors:** Vega González-Bueso, Juan José Santamaría, Ignasi Oliveras, Daniel Fernández, Elena Montero, Marta Baño, Susana Jiménez-Murcia, Amparo del Pino-Gutiérrez, Joan Ribas

**Affiliations:** 1Atención e Investigación en Socioadicciones (AIS), Mental Health and Addictions Network, Generalitat de Catalunya (XHUB), 08014 Barcelona, Spain; 2Department of Psychiatry & Forensic Medicine, Institute of Neurosciences, Universitat Autònoma de Barcelona, Bellaterra, 08193 Barcelona, Spain; 3Research and Development Unit, Parc Sanitari Sant Joan de Déu, Fundació Sant Joan de Déu, Sant Boi de Llobregat, 08830 Barcelona, Spain; 4Department of Statistics and Operations Research, Polytechnic University of Catalonia-BarcelonaTech, 08028 Barcelona, Spain; 5Centro de Investigación Biomédica en Red de Salud Mental (CIBERSAM), Instituto de Salud Carlos III, 28029 Madrid, Spain; 6Pathological Gambling Unit, Department of Psychiatry, Bellvitge University Hospital-IDIBELL, 08907 Barcelona, Spain; 7Ciber Fisiopatología Obesidad y Nutrición (CIBERObn), Instituto de Salud Carlos III, 28029 Madrid, Spain; 8Department of Clinical Sciences, School of Medicine and Health Sciences, University of Barcelona, 08907 Barcelona, Spain; 9Nursing Department of Mental Health, Public Health, Maternal and Child Health, Nursing School of the University of Barcelona, 08907 Barcelona, Spain

**Keywords:** internet gaming disorder, cluster analysis, video game, video game addiction, personality, comorbidity

## Abstract

In recent years, the evidence regarding Internet Gaming Disorder (IGD) suggests that some personality traits are important risk factors for developing this problem. The heterogeneity involved in problematic online gaming and differences found in the literature regarding the comorbid psychopathology associated with the problem could be explained through different types of gamers. Clustering analysis can allow organization of a collection of personality traits into clusters based on similarity. The objectives of this study were: (1) to obtain an empirical classification of IGD patients according to personality variables and (2) to describe the resultant groups in terms of clinical and sociodemographic variables. The sample included 66 IGD adolescent patients who were consecutive referrals at a mental health center in Barcelona, Spain. A Gaussian mixture model cluster analysis was used in order to classify the subjects based on their personality. Two clusters based on personality traits were detected: type I “higher comorbid symptoms” (*n* = 24), and type II “lower comorbid symptoms” (*n* = 42). The type I included higher scores in introversive, inhibited, doleful, unruly, forceful, oppositional, self-demeaning and borderline tendency traits, and lower scores in histrionic, egotistic and conforming traits. The type I obtained higher scores on all the Symptom Check List-90 items-Revised, all the State-Trait Anxiety Index scales, and on the DSM-5 IGD criteria. Differences in personality can be useful in determining clusters with different types of dysfunctionality.

## 1. Introduction

According to the ICD-11 (International Classification of Diseases) [1], Internet Gaming Disorder (IGD) is defined as “a pattern of persistent or recurrent gaming behavior (‘digital gaming’ or ‘video-gaming’), which may be online (i.e., over the Internet) or offline, manifested by: (1) impaired control over gaming (e.g., onset, frequency, intensity, duration, termination, context); (2) increasing priority given to gaming to the extent that gaming takes precedence over other life interests and daily activities; and (3) continuation or escalation of gaming despite the occurrence of negative consequences. The behavior pattern is of sufficient severity to result in significant impairment in personal, family, social, educational, occupational or other important areas of functioning. The pattern of gaming behavior may be either continuous or, on the other hand, episodic and recurrent. The gaming behavior and other features are normally evident over a period of at least 12 months for a diagnosis to be assigned, although the required duration may be shortened if all diagnostic requirements are met and symptoms are severe”.

Nowadays, behavioral addictions, including IGD, are increasingly being documented worldwide [2]. The current versions of the official diagnostic classification manuals have included addictions without substances in the behavioral addictions category. At the moment, only Gambling Disorder has been included in this category, and although IGD seems to share many factors with this disorder such as the negative reinforcement as a maintaining variable in the long-term maintenance of the behavior, or the use of positive reinforcement as a developing mechanism at the beginning of the problem [3,4], the DSM-5 work group decided to include IGD in Section III of the diagnostic manual DSM-5 [5] as a condition that requires further study.

In recent years, numerous factors involved in the etiology of problematic online gaming and of IGD have been identified. Although more research is needed, the evidence suggests that some personality traits are important risk factors for developing this problem [6,7]. Personality traits are different across individuals, influence their behavior, reflect people’s characteristic patterns of thoughts, attitudes, emotions, and behaviors [8], and they have relatively high stability over a lifetime, even though some authors have found changes after cognitive behavioral therapies [9]. Specifically, researchers have linked patients with IGD to personality traits common in other addictive disorders.

Regarding the Big Five Model of personality, some of the factors of the model seem to have relevance in the addiction process [10]. Among them, high neuroticism is the most commonly present in IGD, and has been interpreted as a way to use online gaming to overcome negative life effects [11] or to modify negative emotions [12,13]. The results regarding the other traits of the model and male gamers are various, with low agreeableness [14], low conscientiousness and low extraversion [15] being the most consistent findings. These domains reflect an impaired offline social cooperation with their peers, low self-discipline, and low motivation to maintain positive interpersonal relations.

Taking other personality models as a reference, some authors have found that low sociability [16], low openness to experience [17], and the combination of low self-directedness and cooperativeness [18] were correlated with problematic gaming and IGD. Finally, low self-directness [14] has been found as a predictor of IGD. This trait reflects the lack of regulation and adaptation of one’s own behavior in order to achieve personally chosen goals and values, and the tendency to be laid back.

All these traits combined can predispose people to avoid social interactions and new activities. This diminished social participation in the “real world” and the need to establish some kind of social interactions with others could lead gamers to find online relationships, which are commonly more distant and superficial, with people with similar interests. Thus, they expand their social network [19] and find themselves in a safe environment when they are in the online world.

Clustering can be defined as the statistical methods that allow us to organize a collection of data points into clusters based on similarity [20]. These methods are part of what we call unsupervised learning and they are used in a wide range of research fields such as psychology, biology and market research [21,22,23]. Clustering methods have been used to classify individuals based upon behavioral patterns [24], personality traits [25] and severity of mental disorder [26].

Several studies have focused on classifying the personality traits of patients with behavioral addictions using clusters, and their relationships with psychiatric comorbidity and sociodemographic characteristics. The study of personality profiles for gambling disorder [25,27,28] and compulsive buying [29] shows that there is a heterogeneity in the personality patterns of the affected people, and these different profiles are associated with differences in the number and severity of other psychological comorbid symptoms. These results suggest that the experience with problematic behavior varies between patients, and that the processing of the negative consequences derived from it may affect them in different ways.

Focusing on the use of videogames, Billieux et al. [30] classified a group of 1057 general population online gamers, playing a massive multiplayer online roleplaying-game, into five reliable clusters (three problematic and two non-problematic clusters) according to several psychological risk factors (impulsivity, motives to play, self-esteem) and potential consequences of playing (addiction symptoms, positive and negative affect). Members of the two non-problematic clusters were defined by low impulsivity and high and low levels of self-esteem, respectively. The motivations to play of these two clusters vary between non-fundamental, to motivations related to social exchange and roleplaying.

With respect to the three problematic clusters, the first one was composed of gamers with poor self-esteem and high impulsivity (but low sensation seeking), and by low achievement and high escapism motives. The second problematic cluster included high self-esteem, high impulsivity, and motivations regarding achievements in the game. The last cluster comprise gamers with high self-esteem, high impulsivity, and motives related to roleplaying, achievements and escapism. This study demonstrated the existence of distinct subtypes of problematic online gamers, emphasizing the high heterogeneity and the wide range of psychological factors involved in the problem.

Therefore, it could be possible that, similar to what happens in other behavioral addictions, the different personality types of the gamers have a role not only in the development or perpetuation of the problem, but also in the comorbid psychopathology associated to the disorder. In that sense, several authors have analyzed the different personality and psychopathological features among IGD patients [31,32], finding fewer functional personality traits and higher psychopathological scores compared with a normative population.

Nevertheless, the relationship between personality and psychopathology in IGD remains unclear, and the associations between comorbidity and IGD in adolescents and young adults have shown inconsistent results with depression, anxiety, ADHD or hyperactivity, social phobia/anxiety, and obsessive-compulsive disorder, finding among different authors full, partial and no associations with these symptoms [33]. It is possible that these inconsistencies could be related to the analysis as a whole of different personality types among video game players.

In other words, cluster analyses can help in the conceptualization of patients consulting for IGD, and these profiles of similarities and differences among individuals can contribute to clarifying some of the results found in previous research regarding clinical profiles, and can help to improve the clinical treatments. Considering this, the objectives of this study are as follows: (1) to obtain an empirical classification of IGD patients according to personality variables and (2) to describe the resultant groups in terms of clinical and sociodemographic variables.

## 2. Materials and Methods

### 2.1. Participants

The sample included 66 IGD patients who were consecutive referrals for assessment and outpatient treatment at the Behavioral Addiction Unit in the mental health center AIS-PRO JUVENTUD (Care and Research in Behavioral Addiction) (AIS), located in Barcelona, Spain.

The required sample size was calculated based on the standard deviations of the questionnaire Millon Adolescent Personality Inventory (MACI). Thus, by setting an alpha risk of 0.05 and a beta risk of 0.20 in a two-sided test with a 10% estimated dropout rate, we required a sample size of 59 individuals to detect a minimum expected difference between groups of 6 units. We therefore decided to recruit 66 patients.

The exclusion criteria for being included in the analyses were: (1) had neurological disorders or primary psychiatric conditions that could affect cognitive function (assessed through semi-structured, face-to-face, clinical interview, (2) had a head injury with loss of consciousness for more than 2 min or a learning disorder, (3) used psychostimulants or drugs that could interfere with the assessment, (4) were older than 21 years or younger than 12. No potential participants were excluded based on exclusion criteria 1, 2, or 3.

This study was carried out according to the latest version of the Declaration of Helsinki. The Ethics Committee of CEIC Fundació Unió Catalana d’Hospitals (CEIC14/71) approved the study, and informed consent (signed document) was obtained from parents of adolescents under the age of 18 years and adolescents over the age of 18 years (and assent in adolescents under the age of 18 years).

The characteristics of the sample were as follows: all patients were Caucasian and male. All the adolescents included have as their main and problematic videogame an online videogame, with a mean age of 15.80 (SD 2.18) years. Most patients had elementary education (92.4%). Regarding their main problem, all the patients included played online videogames, and the mean duration of the problem was 2.2 (SD 1.7) years. There was no consumption of alcohol or drugs and only a 3.0% of the sample were smokers.

### 2.2. Instruments

#### 2.2.1. Millon Adolescent Personality Inventory (MACI) 

This personality test [34] has 160 items and is self-administered. It measures thirty-one scales: twelve Personality Patterns scales (Axis II), eight Expressed Concerns Scales, seven Clinical Syndrome Scales, three Modifying Indices (particular response styles), and a Validity scale. The instrument has been translated and validated into a Spanish population with a good internal consistency of 0.82 (mean Cronbach’s alpha) [35]. The MACI is one of the most widely used personality assessment tests for adolescents [36,37,38,39,40]. The MACI is constructed using an underlying theory of personality and psychopathology, and can identify and assess a wide range of psychological difficulties in adolescents. Studies have examined the potential utility of the MACI for assessing substance use disorders [41], reporting support for MACI as a screening instrument. We used this instrument in order to facilitate current and future comparisons in other studies regarding IGD or other psychological problems.

#### 2.2.2. Symptom CheckList-90 Items-Revised (SCL-90-R) 

This questionnaire [42] evaluates psychological problems and psychopathological symptoms. It contains ninety items and measures nine primary symptom dimensions: somatization, obsession-compulsion, interpersonal sensitivity, depression, anxiety, hostility, phobic anxiety, paranoid ideation, and psychoticism. It also includes three global indices, such as a global severity index (GSI), that measures overall psychological distress, a positive symptom distress index (PSDI), to measure the intensity of symptoms; and a positive symptom total (PST). This scale has been translated to Spanish and validated in a Spanish population [43], and presents a good internal consistency (mean Cronbach’s alpha = 0.75).

#### 2.2.3. State-Trait Anxiety Index (STAI) 

This questionnaire [44] includes forty items on a 4-point rating scale and is self-reported, measuring state anxiety (twenty items) and trait anxiety (twenty items). The minimum score is 20 and the maximum is 80 points. The state anxiety uses items that measure subjective feelings of apprehension, tension, nervousness, worry, and activation/arousal of the autonomic nervous system and evaluates the current state of anxiety. The trait anxiety scale includes general states of calmness, confidence, and security and evaluates relatively stable aspects of “anxiety proneness”. The STAI has been translated to Spanish and validated in the Spanish population with a mean Cronbach’s alpha coefficient of 0.92 [45].

#### 2.2.4. DSM-5 IGD Criteria 

This instrument is a questionnaire evaluating the criteria for IGD proposed in the DSM-5 [5]. The diagnostic criteria of IGD are composed of 9 items: preoccupation, withdrawal, tolerance, unsuccessful attempts to control, loss of other interests, continued excessive use despite psychosocial problems, deceiving regarding online gaming, escape, and functional impairment. Five or more DSM-5 criteria of IGD indicates Internet gaming problems. The criteria were asked in a questionnaire form using a “yes” or “no” response.

#### 2.2.5. Sociodemographical Variables

Additional demographic, clinical, and social/family variables related to internet gaming were measured using a semi-structured face-to-face clinical interview, including age, sex, duration of the problem, and education level.

### 2.3. Procedure

First, at intake, a face-to-face specific clinical interview and a functional analysis of IGD was carried by experienced psychologists (more than 5 years of clinical experience in behavioral addictions) using the semi-structured clinical interview SCID-I [46]. The questions included in this interview were about tolerance, preoccupation, withdrawal, loss of control, playing for long periods, escaping from adverse mental states, risking or losing relationships or opportunities because of the behavior, deception/covering up, giving up other activities, persistence of the behavior despite problems, and functional impairment (e.g., functional impairment in familial relationships, other social relationships, and academic achievement), and questions regarding demographic data.

During a second session (with an average duration of 90 min) before starting the treatment, were administered the above-mentioned questionnaires.

With regard to meeting the diagnostic criteria for IGD, the results obtained through the DSM-5 diagnostic criteria questionnaire were compared post hoc with the results obtained through the face-to-face clinical interview, and only patients who met the DSM-5 criteria for IGD were included in our analysis.

### 2.4. Statistical Analysis

Statistical analysis was carried out using the statistical software R version 3.5.3 (R Core Team, Vienna, Austria) and, in particular, its R packages mclust [47] and factoextra [48].

We obtained clusters based on the scores of the 12 sub-scales of the personality patterns scale of the MACI (Introversive, Inhibited, Doleful, Submissive, Histrionic, Egotistic, Unruly, Forceful, Conforming, Oppositional, Self-Demeaning, and Borderline Tendency) conducting a Gaussian Mixture Model (GMM) cluster analysis. Model-fitting is performed using the expectation-maximization (EM) algorithm [49,50], which later is used to initialize a hierarchical model-based agglomerative clustering [51,52]. The optimal number of clusters was selected based on the Bayesian Information Criterion (BIC), where the lowest values indicate better fit. Each individual was allocated to one class only according to their highest probability of membership.

Chi-square tests (χ^2^) for categorical variables and *t*-test (2 groups) for quantitative measures were computed to assess differences between clusters. When the normality assumption was not accomplished according to Shapiro–Wilk test, the equivalent non-parametric Mann–Whitney U test was performed. A two-sided *p*-value < 0.05 was considered statistically significant. Cohen’s *d* was used to measure the effect size for power analysis purposes. The effect size was classified as high (*d* = 0.8), medium (*d* = 0.5) or low (*d* = 0.2) according to Cohen [53].

## 3. Results

### 3.1. Cluster Composition: Description for the Cluster Indicators

In the MACI scale (*n* = 66 men, age: 15.80 ± 2.18) there are 12 sub-scales: Introversive, Inhibited, Doleful, Submissive, Histrionic, Egotistic, Unruly, Forceful, Conforming, Oppositional, Self-Demeaning, and Borderline Tendency. The descriptive statistics of the sample are presented in Table 1.

We conducted a Gaussian finite mixture model cluster analysis. The results, for the personality patterns scale, showed that a 2 cluster (VEE: ellipsoidal, equal shape and orientation) was the optimal solution. The cluster classification for each subject is shown in Figure 1. The first cluster has 24 subjects and the second cluster has 42 subjects.

The results of the *t*-test and the Mann–Whitney U test comparing the two clusters created by the model-based clustering algorithm show that there are significant differences in all the sub-scales, except in the Submissive. The first cluster has higher values in the following sub-scales: introversive, inhibited, doleful, unruly, forceful, oppositional, self-demeaning and borderline tendency, while in the histrionic, egotistic and conforming sub-scales the first cluster has lower values (see Table 2 and Table 3). The results of the Cohen’s d show that the effect size was high in all scales apart from the Submissive and Unruly scales (*d*-values < 0.8; Table 2 and Table 3).

### 3.2. Comparison between the Clusters in Sociodemographic and Clinical Variables

After that, we compared the two clusters regarding a series of sociodemographic variables, i.e., age, age of onset of the disorder, disorder duration, and education level. The results of the *t*-tests showed that there were no significant differences between clusters regarding age, age of onset, and years with IGD. Besides, the Fisher exact test, to elucidate whether the education level was related to the cluster classification, was also non-significant.

In order to investigate if there were other significant psychopathological differences between clusters, we conducted Student’s *t*-tests and Mann–Whitney U-tests to compare the clusters in the Symptom Check List-90 items-Revised scores, in the DSM 5 diagnostic criteria scores and the STAI. The SCL-R has nine dimensions: somatization, obsessive-compulsive, interpersonal sensitivity, depression, anxiety, hostility, phobic anxiety, paranoid ideation and psychoticism. The results of the analyses showed that the first cluster had significantly increased scores in all dimensions of the SCL-R, except for the somatization. Regarding the diagnostic criteria, the analysis comparing the cluster regarding the DSM scores yielded a significant result (U = 684, *p* < 0.05) indicating that the subjects in the first cluster had higher scores. Regarding Cohen’s d values, we observed high effect sizes except for the three dimensions of the SCL-90-R (Somatization, Obsessive-Compulsive and Phobia), and for the DSM-5 scores, the effect size was moderate.

In a similar manner, we conducted the same analyses for the anxiety-trait and anxiety-state of the STAI and all the comparisons were statistically significant (all *t* = 5.54; *p* < 0.001 for anxiety trait and *t* = 3.58; *p* < 0.001 for anxiety state), indicating that the first cluster of each scale had higher anxiety levels than the members of the second cluster.

## 4. Discussion

This work explores the existence of empirical clusters for IGD starting from the patients’ personality trait scores, using a person-centered methodology.

Two clusters were detected: type I “higher comorbid symptoms” and type II “lower comorbid symptoms”. The main differences in the clustering variables between both clusters were the mean scores in the personality traits introversive, inhibited, doleful, unruly, forceful, oppositional, self-demeaning and borderline tendency (higher scores for the type I), histrionic, egotistic and conforming (lower scores for the type I). It should be noted the similarity between this cluster I and the IGD dimensional personality profile found in other studies.

In line with our results, literature about personality traits and IGD suggests that high scores in neuroticism [54], impulsivity [55] and introversion [15,18], and lower levels in agreeableness [14,15], cooperativeness [18] and self-regulation [56], are risk factors for this disorder. Furthermore, some authors have found that low levels of self- directedness uniquely predicted video game abuse in an adult population with Gambling Disorder [14], and others have identified lower responsibility as a risk factor associated with this IGD [57]. Summarizing the conclusions of this previous research, this combination of personality traits could lead the affected people to use online gaming as a maladaptive mood modifier and/or a strategy to overcome negative life events, to having higher tendency toward competition, lower life satisfaction, lower expectations of self-efficacy, less face-to-face social support, and increased feelings of anger.

Analyzing the comorbidity level of both clusters, the type I has the higher comorbidity, obtaining higher scores on all SCL-90R scales (i.e., somatization, obsessive-compulsive, sensitivity, depression, anxiety, hostility, phobia, paranoid ideation and psychoticism) all STAI scales (anxiety trait and state), and on the IGD diagnostic criteria, being the obsessive-compulsive, sensitivity, depression, anxiety, hostility, paranoid ideation, psychoticism and anxiety trait above general population scoring. Despite the literature regarding comorbid psychopathology in adolescents and young adults with IGD having found full associations with anxiety [55], depression [58], ADHD or hyperactivity symptoms [59], and social phobia/anxiety [60], other authors have found partial associations [61,62] and no associations [63] with these same symptoms. These contradictory results and a possible publication bias [33] makes it difficult to detect the directionality and the relationship of these associations and shows the existing complexity of the relationship between IGD, personality and psychopathology.

Some authors have found relationships between personality factors and psychopathology in adolescents. According to the results found by Castellanos-Ryan [64], general psychopathology was related to high disinhibition/impulsivity, low agreeableness, high neuroticism and hopelessness, high delay-discounting and poor response inhibition. Of all of this, high neuroticism is the Big Five trait most strongly associated with several psychopathological symptoms, especially with anxiety and mood disorders and their comorbidity [65]. It has been hypothesized that the negative emotionality related with this trait could mediate this association [66]. Other personality trait predictors of psychopathology in adolescents are low conscientiousness, low extraversion [67], hopelessness and high impulsivity [68]. Regarding the Egotistic personality scale, where cluster 2 scored higher, adolescents who score high in the Egotistic Personality subscale have a passive-independent pattern, are perceived as conceited, have strong self-esteem, may take others for granted, and may fantasize about future success and power [69]. Some authors showed that certain narcissistic features are adaptive when paired with high levels of self-esteem [70]. Taking these results into account, the differences between the Type I and Type II personality traits could suggest that the higher negative affect, the lower positive affect, the difficulties in establishing social support, and the poor general behavioral control involved in the Type I personality profiles may be characterizing the chances of developing comorbid psychopathology, and could partially explain the inconsistent results regarding IGD and psychopathology.

When interpreting the results of this study, the effects of maturity and personality development in adolescence must be taken into account. Recent research has shown a spontaneous recovery of video game addiction [71,72] and it must be noted that personality traits may change during individual development [73]. Therefore, it is possible that the dysfunctional personality traits found will change or disappear in the future, and with them the influence they are having on the development or maintenance of IGD and the associated comorbidity.

In addition, our results show that there is a group of patients with IGD whose comorbid mental health symptoms are non-existent, or similar to the non-psychiatric population. Since there are no differences in the age at which the disorder begins or in the duration of the disorder between both cluster groups, it seems that in at least some individuals, IGD presents itself as a solitary condition or as an originating condition.

Regardless of the debate about the diagnosis and comorbidity [74], across the years the prevalence of the disordered gaming and the incidence of patients seeking treatment for IGD has remained stable [75]. This study is focused on the heterogeneity of the disorder and in the existence of different subgroups of IGD patients, based on personality, with differences in the seriousness of their psychological comorbidity. Our results suggest that these variables appear to be useful in determining clusters which represent different clinical subtypes with different degrees of severity. Such differences among online gamers imply that the experience of playing may vary in patients and the general population. Therefore, in order to succeed in developing instruments, planning efficient prevention programs for general population and targeting at-risk gamers, these dependent effects should be considered. Furthermore, the data regarding standard therapy for IGD are limited, with cognitive-behavioral therapy, family therapy and pharmacological intervention showing some significant results [76,77]. In order to develop more specific and accurate treatment interventions, the existence of IGD clusters should be considered.

Concerning limitations, the results of this study are based on a small sample, but considering that the sample is formed by a clinical sample, evaluated in a controlled setting, there may be a reason to anticipate that our findings may be of value for similar profile patients. Second, the measures used are based on self-administered questionnaires, although a trained psychologist supervised the entire process. Third, the cross-sectional design does not allow us to determine causality of the variables assessed. Fourth, the severity of IGD has not been reflected in this paper, as the objective of this work was to describe the different types of IGD-diagnosed adolescents with respect to personality traits and to analyze the comorbidity of each cluster groups.

Future research could complement these results using longitudinal designs addressing the potential mediating role of personality in the etiological factors and clinical course of IGD, in order to try to develop a predictive model of problematic video games use and its comorbidity.

## 5. Conclusions

Problematic use of online videogames is a growing clinical issue in developed countries. There is ongoing debate about the suitability of the current proposed definition and diagnostic criteria and about its related dysfunctional consequences and clinical correlates. The current literature shows that IGD is a complex problem where neurobiological systems, personality, and socioeconomic and environmental variables are involved. The objective of this study was to explore the heterogeneity of the affected people and the existence of differentiated subgroups based on different personality traits.

The results suggest that differences in personality can be useful in determining clusters with different levels of comorbidity. These differences imply that the experience of playing online videogames may affect different players, and clinicians and researchers should consider them in order to develop treatment options or assessment instruments. Future research should include more heterogeneous samples in relation to age and sex, and consider treatment outcome as a variable to analyze.

## Figures and Tables

**Figure 1 ijerph-17-01516-f001:**
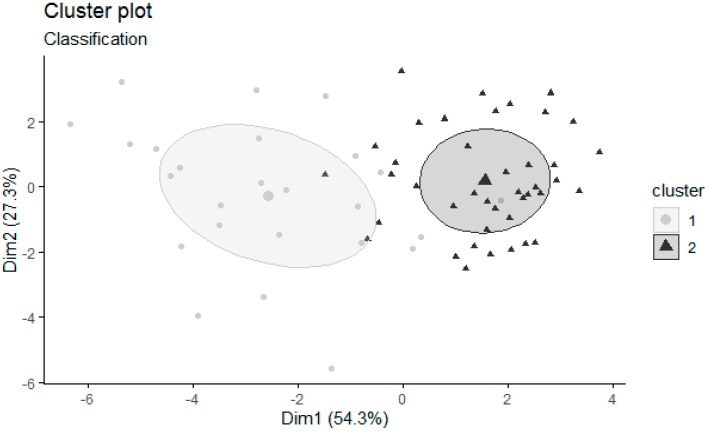
Classification of subjects for each cluster.

**Table 1 ijerph-17-01516-t001:** Descriptive statistics of the personality traits of the Millon Adolescent Personality Inventory (MACI) scale (*n* = 66).

Personality Traits	Mean	SD	Median	IQR	Min	Max
Introversive	25.20	11.61	23	17.80	9	52
Inhibited	20.86	10.82	18.50	25.50	4	48
Doleful	11.47	9.92	8	14	0	41
Submissive	43.85	10.00	45.50	13	21	69
Histrionic	36.55	9.97	37	14.30	12	54
Egotistic	33.47	10.85	35	15	3	51
Unruly	30.23	9.39	29.50	11.80	10	52
Forceful	10.59	6.81	9	7.6	0	34
Conforming	45.41	9.05	46	13.30	18	62
Oppositional	20.30	9.86	20	12.80	4	44
Self-Demeaning	19.42	13.84	16	20	0	55
Borderline Tendency	11.70	7.72	11	12.50	0	30

SD = Standard Deviation, IQR = Interquartile range.

**Table 2 ijerph-17-01516-t002:** T-test comparison between clusters of the personality traits of the MACI and the State-Trait Anxiety Index (STAI) scales (cluster 1 *n* = 24, cluster 2 *n* = 42).

Independent Samples Test						
	CLUSTER 1	CLUSTER 2				
	Mean (SD)	Mean (SD)	t	df	Sig. (2 tailed)	Cohen’s d
MACI						
Submissive	46.48 (11.86)	42.24 (8.42)	1.69	64	0.126	0.43
Egotistic	27.80 (3.03)	36.93 (7.33)	−3.16	33.15	0.003	**−0.92**
Unruly	34.12 (9.70)	27.85 (8.45)	2.76	64	0.010	0.70
Conforming	39.44 (8.19)	49.04 (7.49)	−4.86	64	0.000	**−1.23**
Oppositional	28.76 (8.58)	15.14 (6.46)	7.32	64	0.000	**1.86**
STAI						
Anxiety State	18.72 (8.35)	11.38 (7.52)	3.67	63	0.000	**0.94**
Anxiety Trait	23.48 (8.11)	12.64 (7.00)	5.75	64	0.000	**1.46**

SD = Standard Deviation; df = degrees of freedom (Cohen’s d > 0.80 in bold).

**Table 3 ijerph-17-01516-t003:** Mann–Whitney U-test for the MACI, Symptom CheckList-90 items-Revised (SCL-90-R), and DSM 5 criteria mean scores (cluster 1 *n* = 24, cluster 2 *n* = 42).

Mann–Whitney U Test					
	CLUSTER 1	CLUSTER 2			
	Mean (SD)	Mean (SD)	U	Sig. (2 tailed)	Cohen’s d
MACI					
Introversive	35.76 (9.27)	18.76 (7.44)	948.50	0.000	**2.08**
Inhibited	29.88 (11.04)	15.37 (5.94)	883	0.000	**1.76**
Doleful	20.44 (9.42)	6.00 (5.07)	941	0.000	**2.05**
Histrionic	29.88 (10.95)	40.61 (6.70)	223.50	0.000	**−1.26**
Forceful	14.56 (7.49)	8.17 (5.07)	781	0.000	**1.05**
Self-Demeaning	32.96 (11.51)	11.17 (6.97)	974	0.000	**2.45**
Borderline Tend.	19.44 (5.34)	6.97 (4.39)	998.5	0.000	**2.61**
SCL-90-R					
Somatization	0.55 (0.54)	0.26 (0.27)	718	0.006	0.75
Obsessive-comp	1.03 (0.60)	0.61 (0.51)	719	0.006	0.76
Interp. sens.	1.00 (0.81)	0.38 (0.39)	781	0.000	**1.04**
Depression	0.93 (0.72)	0.28 (0.30)	840.50	0.000	**1.28**
Anxiety	0.69 (0.81)	0.21 (0.22)	700	0.012	**0.90**
Hostility	1.10 (0.91)	0.56 (0.45)	708	0.009	**0.81**
Phobia	0.40 (0.57)	0.10 (0.19)	782.50	0.000	0.77
Paranoid ideation	1.03 (0.82)	0.38 (0.43)	773	0.000	**1.06**
Psychoticism	0.56 (0.50)	0.14 (0.20)	845	0.000	**1.18**
Global severity	0.80(0.56)	0.33(0.25)	833	0.000	**1.20**
DSM 5 criteria	5.84 (1.82)	4.95 (1.55)	684	0.019	0.54

SD = Standard Deviation; df = degrees of freedom (Cohen’s d > 0.80 in bold).

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
