# Peer review of "Internet Gaming Disorder Clustering Based on Personality Traits in Adolescents, and Its Relation with Comorbid Psychological Symptoms"

_ijerph, 2020, doi:10.3390/ijerph17051516_

Round 1
Reviewer 1 Report
The article is a contribution to the further study of IGD as recommended by the DSM V. It does not tackle the question of whether IGD is or is not a correct designation for the phenomenon identified, but rather brackets that question to investigate the personality types of people who are referred to clinics for IGD. This approach could potentially lead to a better understanding of the variability of personality types within populations diagnosed with IGD, which in turn could help further our understanding of IGD as a diagnosis. The article employs cluster analysis, finding two clusters representing different personality types. The two clusters differed on measures of psychopathology, anxiety, and IGD (as measured by the DSM V criteria), with the first cluster showing higher measures on all three. The results generally support results from previous studies. Regarding the following statement in the Discussions section: “This combination of personality traits leads that affected people to use online gaming as a maladaptive mood modifier and/or a strategy to overcome negative life events, to having higher tendency toward competition, lower life satisfaction, lower expectations of self-efficacy, less face-to-face social support, and increased anger feelings.” It is not clear if this is a summary of previous research or an assertion based on the current findings. My sense is that it is a summary of previous research, and so a clearer indication of this would be appreciated. There is nothing in the current study that would serve as evidence for this assertion, since the study cannot (as the authors acknowledge) show causality. The discussion helpfully points to the complexity of IGD as a diagnosis, for example in direction of causality and comorbidity. An important discussion of comorbiditty in IGD can be found in Bean, A. M., Nielsen, R. K. L., Van Rooij, A. J., & Ferguson, C. J. (2017). Video Game Addiction: The Push to Pathologize Video Games. Professional Psychology: Research and Practice, 48(5), 378–389. https://doi.org/10.1037/pro0000150. The current findings show that people who score highest on IGD also score highest on anxiety and psychopathology--I feel that there is a lot to be unpacked here in the discussion section regarding the nature of IGD and its limitations as a diagnosis. The discussion section seems quite slight to me. Therefore, my main criticism of the article is with regard to the significance of the findings, and how that significance is communicated in the discussion section. The results need to be further unpacked to demonstrate how these results further our understanding of IGD. The writing is generally clear, but there are a number of grammatical errors and so I would recommend a professional proof-reader go over the article before publication.Author Response
Comments and Suggestions for Authors
The article is a contribution to the further study of IGD as recommended by the DSM V. It does not tackle the question of whether IGD is or is not a correct designation for the phenomenon identified, but rather brackets that question to investigate the personality types of people who are referred to clinics for IGD. This approach could potentially lead to a better understanding of the variability of personality types within populations diagnosed with IGD, which in turn could help further our understanding of IGD as a diagnosis.
The article employs cluster analysis, finding two clusters representing different personality types. The two clusters differed on measures of psychopathology, anxiety, and IGD (as measured by the DSM V criteria), with the first cluster showing higher measures on all three.
The results generally support results from previous studies.
Regarding the following statement in the Discussions section: “This combination of personality traits leads that affected people to use online gaming as a maladaptive mood modifier and/or a strategy to overcome negative life events, to having higher tendency toward competition, lower life satisfaction, lower expectations of self-efficacy, less face-to-face social support, and increased anger feelings.” It is not clear if this is a summary of previous research or an assertion based on the current findings. My sense is that it is a summary of previous research, and so a clearer indication of this would be appreciated. There is nothing in the current study that would serve as evidence for this assertion, since the study cannot (as the authors acknowledge) show causality.
ANSWER:
Indeed, this paragraph try to summarize the conclusions of this previous research that is in line with our results.
We have added the following sentence to this paragraph:
Summarizing the conclusions of this previous research,
In addition, in order to be more prudent, we have changed “leads” with “could lead”.
The discussion helpfully points to the complexity of IGD as a diagnosis, for example in direction of causality and comorbidity. An important discussion of comorbiditty in IGD can be found in Bean, A. M., Nielsen, R. K. L., Van Rooij, A. J., & Ferguson, C. J. (2017). Video Game Addiction: The Push to Pathologize Video Games. Professional Psychology: Research and Practice, 48(5), 378–389. https://doi.org/10.1037/pro0000150. The current findings show that people who score highest on IGD also score highest on anxiety and psychopathology- I feel that there is a lot to be unpacked here in the discussion section regarding the nature of IGD and its limitations as a diagnosis. The discussion section seems quite slight to me. Therefore, my main criticism of the article is with regard to the significance of the findings, and how that significance is communicated in the discussion section. The results need to be further unpacked to demonstrate how these results further our understanding of IGD.
ANSWER:
We have added the following paragraphs to the Discussion section, emphasizing the significance of the findings:
“In addition, our results show that there is a group of patients with IGD whose comorbid mental health symptoms are non-existent, or similar to the non-psychiatric population. Since there are no differences in the age at which the disorder begins or in the duration of the disorder between both cluster groups, it seems that in at least some individuals, the IGD presents itself as a solitary condition or as an originating condition.”
“Regardless of the debate about the diagnosis and comorbidity [67], across the years the prevalence of the disordered gaming and the incidence of patients seeking treatment for IGD has remained stable [68]. This study is focused on the heterogeneity of the disorder and in the existence of different subgroups of IGD patients, based on personality, with differences in the seriousness of their psychological comorbidity. Our results suggest that these variables appears to be useful in determining clusters, which represent different clinical subtypes with different degrees of severity. Such differences among online gamers imply that the experience of playing may vary between patients and general population. Therefore, in order to get success in developing instruments, planning efficient prevention programs for general population and targeting at-risk gamers these dependent effects should be in consideration. Furthermore, the data regarding standard therapy for IGD is limited, having the cognitive-behavioral therapy, the family therapy and the pharmacological intervention some significant results [69,70]. In order to develop more specific and accurate treatment interventions, the existence of IGD cluster should be considered.”
Bean, A. M.; Nielsen, R. K. L.; van Rooij, A. J.; Ferguson, C. J. Video game addiction: The push to pathologize video games. Prof. Psychol. Res. Pract. 2017, 48, 378–389, doi:10.1037/pro0000150. Feng, W.; E. Ramo, D.; Steven R., C.; A. Bourgeois, J. Internet gaming disorder: Trends in prevalence 1998–2016. Addict. Behav. 2017, 75, 17–24, doi:10.1016/J.ADDBEH.2017.06.010. Han, D. H.; Kim, S. M.; Lee, Y. S.; Renshaw, P. F. The effect of family therapy on the changes in the severity of on-line game play and brain activity in adolescents with on-line game addiction. Psychiatry Res. Neuroimaging 2012, 202, 126–131, doi:10.1016/j.pscychresns.2012.02.011. Kim, S. M.; Han, D. H.; Lee, Y. S.; Renshaw, P. F. Combined cognitive behavioral therapy and bupropion for the treatment of problematic on-line game play in adolescents with major depressive disorder. Comput. Human Behav. 2012, 28, 1954–1959, doi:10.1016/j.chb.2012.05.015.
The writing is generally clear, but there are a number of grammatical errors and so I would recommend a professional proof-reader go over the article before publication.
ANSWER:
Thank you for your advice regarding the English language in the paper, this paper (if accepted), will be included in a Special Issue, and the journal kindly offers a free English editing after the acceptance.
Reviewer 2 Report
This is an interesting study, the researchers assessed IGD patients and classified them with several questionnaires. Two clusters were found in this study, one is related to more severe mental health problem and introversive personality, another is related to less severe mental health. Despite significant results were found, some comments are provided to be considered as followed:
It is good to diagnose the participants, but self-reported IGD still need to be completed. Despite 5-criteria score was applied in this study, but the severity of IDG could not be clearly reflected in this study. You findings supported the relationship between negative personality and IDG. However, we could not understand the differences between most dysfunction and less function group. The section of Introduction, the rationale is unclear. I could not understand clearly, for example, line 58-65. Please to rewrite the section of introduction. did this study focus on video gaming or internet gaming? Please to clarify it. It is good to provide the effect size in this study, but I could not understand why the researchers adopted MACI to assess the levels of personality trait? Please to illustrate more. Despite the participants in most dysfunctional group showed higher negative personality, but the extent of egoistic is higher in less dysfunction group. Please to give us more explanations. Last, please to English edition on this manuscript.Author Response
This is an interesting study, the researchers assessed IGD patients and classified them with several questionnaires. Two clusters were found in this study, one is related to more severe mental health problem and introversive personality, another is related to less severe mental health. Despite significant results were found, some comments are provided to be considered as followed:
1- It is good to diagnose the participants, but self-reported IGD still need to be completed. Despite 5-criteria score was applied in this study, but the severity of IDG could not be clearly reflected in this study.
ANSWER:
We never pretended to reflect the severity of IGD, but to reflect the difference types of IGD diagnosed adolescents with respect to personality traits and to analyze the comorbidity of each cluster groups.
We have reflected this in the limitations of the study:
“Fourth, the severity of the IGD has not been reflected in this paper, the objective of this work was to the difference types of IGD diagnosed adolescents with respect to personality traits and to analyze the comorbidity of each cluster groups.”
2- You findings supported the relationship between negative personality and IDG. However, we could not understand the differences between most dysfunction and less function group.
ANSWER:
When we named the two clusters as “Less dysfunctional” and “Most dysfunctional”, our intention was to reflect the difference between the two groups regarding in the comorbid psychiatric symptoms measured by the SCL-90-R and STAI scales, where the cluster I group “most dysfunctional” scored higher than cluster II in all SCL-90-R and STAI dimensions, and near or over the Centile 50 of psychiatric population in the SCL-90-R scales Obsessive-compulsive, Interpersonal Sensitivity, Hostility, Paranoid ideation and Psychoticism, and in the STAI scale Anxiaty Trait.
To clarify this aspect we have changed the name of both clusters to “higher comorbid symptoms” and “lower comorbid symptoms” and we have added this information in the discussion section:
“Analyzing the comorbidity level of both clusters, the type I has the higher comorbidity, …”
And in the conclusions section:
“The results suggest that the differences in personality can be useful in determining clusters with different levels of comorbidity.”
And we have changed this information in the abstract and in all sections of the paper:
“Two clusters based on personality traits were detected: type I “higher comorbid symptoms”, and type II “lower comorbid symptoms”.
3- The section of Introduction, the rationale is unclear. I could not understand clearly, for example, line 58-65. Please to rewrite the section of introduction.
ANSWER:
We have rewrote this paragraph in the Introduction section.
“Nowadays, behavioral addictions, including IGD, are increasingly being documented worldwide [9]. The current versions of the official diagnostic classification manuals have included the addictions without substance or the behavioral addictions category.
At the moment, only Gambling Disorder has been included in this category, and although IGD seems to share many factors with this disorder, such as the negative reinforcement as a maintaining variable in the long-term maintenance of the behavior, or the use of positive reinforcement as a developing mechanism at the beginning of the problem [10,11], the DSM-5 work group decided to include the IGD in Section III of the diagnostic manual DSM-5 [12] as a condition that requires further study”.
4- Did this study focus on video gaming or internet gaming? Please to clarify it.
ANSWER:
We have clarified this point by adding this information in the Participants section:
“The characteristics of the sample were as follows: all patients were Caucasian and male. All the adolescents included have as their main and problematic videogame an online videogame”.
5- It is good to provide the effect size in this study, but I could not understand why the researchers adopted MACI to assess the levels of personality trait? Please to illustrate more.
ANSWER:
We have added the following information to the Instruments Section:
“The MACI is one of the most widely used personality assessment tests for adolescents [34–38]. The MACI is constructed using an underlying theory of personality and psychopathology, and can identify and assess a wide range of psychological difficulties in adolescents. Studies have examined the potential utility of the MACI for assessing substance use disorders [39], reporting support for MACI as a screening instrument. We used this instrument in order to facilitate current and future comparisons in other studies regarding IGD or other psychological problems.”
Murrie, D. C.; Cornell, D. G. Psychopathy screening of incarcerated juveniles: a comparison of measures. Psychol. Assess. 2002, 14, 390–6. Salekin, R. T.; Larrea, M. A.; Ziegler, T. Relationships between the MACI and the BASC in the assessment of child and adolescent offenders. J. Forensic Psychol. Pract. 2002, 2, 35–50, doi:10.1300/J158v02n04_02. Velting, D. M.; Rathus, J. H.; Miller, A. L. MACI personality scale profiles of depressed adolescent suicide attempters: A pilot study. J. Clin. Psychol. 2000, 56, 1381–1385, doi:10.1002/1097-4679(200010)56:10<1381::AID-JCLP9>3.0.CO;2-R. Burton, D. L.; Duty, K. J.; Leibowitz, G. S. Differences between sexually victimized and nonsexually victimized male adolescent sexual abusers: Developmental antecedents and behavioral comparisons. J. Child Sex. Abus. 2011, 20, 77–93, doi:10.1080/10538712.2011.541010. Ferrer, L.; Kirchner, T. Suicidal tendency among adolescents with adjustment disorder: Risk and protective personality factors. Crisis 2015, 36, 202–210, doi:10.1027/0227-5910/a000309. Grilo, C. M.; Fehon, D. C.; Walker, M.; Martino, S. A comparison of adolescent inpatients with and without wubstance abuse using the Millon Adolescent Clinical Inventory. J. Youth Adolesc. 1996, 25, 379–388, doi:10.1007/BF01537391.
5- Despite the participants in most dysfunctional group showed higher negative personality, but the extent of egoistic is higher in less dysfunction group. Please to give us more explanations.
ANSWER:
We have added the following explanation in the Discussion section:
“Regarding the Egotistic personality scale, where cluster 2 scored higher, adolescent who score high in the Egotistic Personality subscale have Passive-independent pattern; perceived as conceited, have strong self-esteem, may take others for granted, may fantasize about future success and power [65]. Some authors showed that certain narcissistic features are adaptive when paired with high levels of self-esteem [66].”
McCann, J. T. The MACI: Composition and clinical applications. In The Millon inventories: Clinical and personality assessment; Millon, T., Ed.; The Guilford Press, 1997; pp. 363–388. Barry, C. T.; Frick, P. J.; Killian, A. L. The relation of narcissism and self-esteem to conduct problems in children: A preliminary investigation. J. Clin. Child Adolesc. Psychol. 2003, 32, 139–152, doi:10.1207/S15374424JCCP3201_13.
6- Last, please to English edition on this manuscript.
ANSWER:
Thank you for your advice regarding the English language in the paper, this paper (if accepted), will be included in a Special Issue, and the journal kindly offers a free English editing after the acceptance.
Reviewer 3 Report
I have only slight reservations about this paper. The topic is important and up-to-date, the procedure and statistical methods seem to be correct. Another advantage is clarified inclusion and exclusion criteria.
My comments are as follows:
As a measure of effect size, Cohen's d statistic was used, which provides important information in which areas the largest differences between clusters were obtained. First, there is no mention of this analysis in the description of statistical methods. Secondly, the areas where Cohen’s d is the greatest were not indicated when presenting the results Tables 2 and 3 are a little confusing due to the interchangeable use of parametric and nonparametric tests and the proposed layout. In this case, one nonparametric test would be acceptable, moreover the samples size is rather small. Despite the confirmed normal distribution, the differences between mean and median values are large. By the way, there is no annotation in any table about mean values but about mean differences. Shouldn't that be a mean score if it's a median. In table 1 min is always higher than max, and IQR is sometimes wider than the range max/min. Clusters size should be given in the headline of tables and mentioned in the abstract. In lines 174 and 184 one Cronbach value is given, not a range (as in line 193); no comments about the reliability in this sample The Authors have pointed out that the results refer to a clinical sample, but this thread in discussion should be developed (see line 361). Are there any implications for general population, especially youth at the risk of addiction? Reference [24] Journal tile is lacking. Maybe Computers in Human Behavior?Author Response
I have only slight reservations about this paper. The topic is important and up-to-date, the procedure and statistical methods seem to be correct. Another advantage is clarified inclusion and exclusion criteria.
My comments are as follows:
1- As a measure of effect size, Cohen's d statistic was used, which provides important information in which areas the largest differences between clusters were obtained. First, there is no mention of this analysis in the description of statistical methods.
ANSWER:
We have added the following information to the Statistical analysis section:
“Cohen's d was used to measure the effect size for power analysis purposes. The effect size was classified as high (d = 0.8), medium (d = 0.5) or low (d = 0.2) according to Cohen [51].”
Cohen, J. Statistical power analysis for the behavioral sciences; 2nd ed.; L. Erlbaum Associates: Hillsdale N.J., 1988; ISBN 9780805802832.
2- Secondly, the areas where Cohen’s d is the greatest were not indicated when presenting the results
ANSWER:
We have added this information in the Results section:
“The results of the Cohen’s d show that the effect size was high in all scales apart from the Submissive and Unruly scales (d-values < 0.8; Tables 2 and 3) “
And
“Regarding Cohen’s d values we observed high effect sizes except for the three dimensions of the SCL-90-R (Somatization, Obsessive-Compulsive and Phobia) and the DSM-5 scores were the effect size was moderate”.
In addition in Tables 2 and 3 we have marked in bold the d’s above 0.8.
3- Tables 2 and 3 are a little confusing due to the interchangeable use of parametric and nonparametric tests and the proposed layout. In this case, one nonparametric test would be acceptable, moreover the samples size is rather small. Despite the confirmed normal distribution, the differences between mean and median values are large. By the way, there is no annotation in any table about mean values but about mean differences. Shouldn't that be a mean score if it's a median.
ANSWER:
We have separated the results in two tables: Table 2 for t-tests and table 3 for Mann Whitney's U.
Regarding the question about means and medians, the Mann-Whitney test compares the mean ranks and it does not compare medians, so we think that for that reason and for comparing results between Table 2 and 3 is better to keep the means in the results.
4- In table 1 min is always higher than max, and IQR is sometimes wider than the range max/min.
ANSWER:
We have included a correct version of this Table.
5- Clusters size should be given in the headline of tables and mentioned in the abstract.
ANSWER:
The N of the clusters has been added in the abstract and in the tables.
6- In lines 174 and 184 one Cronbach value is given, not a range (as in line 193); no comments about the reliability in this sample
ANSWER:
We have reported the mean Cronbach values for all the questionnaires.
We have added the following information to the Instruments Section:
“The MACI is one of the most widely used personality assessment tests for adolescents [34–38]. The MACI is constructed using an underlying theory of personality and psychopathology, and can identify and assess a wide range of psychological difficulties in adolescents. Studies have examined the potential utility of the MACI for assessing substance use disorders [39], reporting support for MACI as a screening instrument. We used this instrument in order to facilitate current and future comparisons in other studies regarding IGD or other psychological problems.”
Murrie, D. C.; Cornell, D. G. Psychopathy screening of incarcerated juveniles: a comparison of measures. Psychol. Assess. 2002, 14, 390–6. Salekin, R. T.; Larrea, M. A.; Ziegler, T. Relationships between the MACI and the BASC in the assessment of child and adolescent offenders. J. Forensic Psychol. Pract. 2002, 2, 35–50, doi:10.1300/J158v02n04_02. Velting, D. M.; Rathus, J. H.; Miller, A. L. MACI personality scale profiles of depressed adolescent suicide attempters: A pilot study. J. Clin. Psychol. 2000, 56, 1381–1385, doi:10.1002/1097-4679(200010)56:10<1381::AID-JCLP9>3.0.CO;2-R. Burton, D. L.; Duty, K. J.; Leibowitz, G. S. Differences between sexually victimized and nonsexually victimized male adolescent sexual abusers: Developmental antecedents and behavioral comparisons. J. Child Sex. Abus. 2011, 20, 77–93, doi:10.1080/10538712.2011.541010. Ferrer, L.; Kirchner, T. Suicidal tendency among adolescents with adjustment disorder: Risk and protective personality factors. Crisis 2015, 36, 202–210, doi:10.1027/0227-5910/a000309. Grilo, C. M.; Fehon, D. C.; Walker, M.; Martino, S. A comparison of adolescent inpatients with and without wubstance abuse using the Millon Adolescent Clinical Inventory. J. Youth Adolesc. 1996, 25, 379–388, doi:10.1007/BF01537391.
7- The Authors have pointed out that the results refer to a clinical sample, but this thread in discussion should be developed (see line 361). Are there any implications for general population, especially youth at the risk of addiction?
ANSWER:
We have added the following paragraph to the discussion section:
“Regardless of the debate about the diagnosis and comorbidity [67], across the years the prevalence of the disordered gaming and the incidence of patients seeking treatment for IGD has remained stable [68]. This study is focused on the heterogeneity of the disorder and in the existence of different subgroups of IGD patients, based on personality, with differences in the seriousness of their psychological comorbidity. Our results suggest that these variables appears to be useful in determining clusters, which represent different clinical subtypes with different degrees of severity. Such differences among online gamers imply that the experience of playing may vary between patients and general population. Therefore, in order to get success in developing instruments, planning efficient prevention programs for general population and targeting at-risk gamers these dependent effects should be in consideration. Furthermore, the data regarding standard therapy for IGD is limited, having the cognitive-behavioral therapy, the family therapy and the pharmacological intervention some significant results [69,70]. In order to develop more specific and accurate treatment interventions, the existence of IGD cluster should be considered.”
Bean, A. M.; Nielsen, R. K. L.; van Rooij, A. J.; Ferguson, C. J. Video game addiction: The push to pathologize video games. Prof. Psychol. Res. Pract. 2017, 48, 378–389, doi:10.1037/pro0000150. Feng, W.; E. Ramo, D.; Steven R., C.; A. Bourgeois, J. Internet gaming disorder: Trends in prevalence 1998–2016. Addict. Behav. 2017, 75, 17–24, doi:10.1016/J.ADDBEH.2017.06.010. Han, D. H.; Kim, S. M.; Lee, Y. S.; Renshaw, P. F. The effect of family therapy on the changes in the severity of on-line game play and brain activity in adolescents with on-line game addiction. Psychiatry Res. Neuroimaging 2012, 202, 126–131, doi:10.1016/j.pscychresns.2012.02.011. Kim, S. M.; Han, D. H.; Lee, Y. S.; Renshaw, P. F. Combined cognitive behavioral therapy and bupropion for the treatment of problematic on-line game play in adolescents with major depressive disorder. Comput. Human Behav. 2012, 28, 1954–1959, doi:10.1016/j.chb.2012.05.015.
8- Reference [24] Journal tile is lacking. Maybe Computers in Human Behavior?
ANSWER:
We have added the missing information:
Gamito, P. S.; Morais, D. G.; Oliveira, J. G.; Brito, R.; Rosa, P. J.; Gaspar De Matos, M. Frequency is not enough: Patterns of use associated with risk of Internet addiction in Portuguese adolescents. Comput. Human Behav. 2016, 58, 471–478, doi:10.1016/j.chb.2016.01.013.
Round 2
Reviewer 1 Report
I am happy that the changes to the manuscript make it an interesting and useful addition to the debate on this topic.
Author Response
Thank you for your suggestions to improve the manuscript.
Reviewer 2 Report
thanks for the authors' revision, this manuscript is closer to be published.
Only two suggestions are provided to be considered, one is a gap from line 132 to 141, please to rewrite. The authors tried to focus on the commorbidity of IGD from the effects of personality trait on IGD. It is still some inconsistent. The authors cited the references about IGD, personality and mental illness, the references are listed as followed:
1.Torres-Rodríguez, A., Griffiths, M. D., Carbonell, X., & Oberst, U. (2018). Internet gaming disorder in adolescence: Psychological characteristics of a clinical sample. Journal of behavioral addictions, 7(3), 707-718.
2.Mallorqui-Bague, N., Fernandez-Aranda, F., Lozano-Madrid, M., Granero, R., Mestre-Bach, G., Bano, M., ... & Jimenez-Murcia, S. (2017). Internet gaming disorder and online gambling disorder: clinical and personality correlates. Journal of behavioral addictions, 6(4), 669-677.
Another is the effect of aging. Actually, the maturity and development of personality during adolescence should be considered. It could be conservative to discuss the effect of personality on IGD during adolescence.
Thanks for the authors' effort and revision.
Author Response
Thanks for the authors' revision, this manuscript is closer to be published.
Only two suggestions are provided to be considered, one is a gap from line 132 to 141, please to rewrite. The authors tried to focus on the commorbidity of IGD from the effects of personality trait on IGD. It is still some inconsistent. The authors cited the references about IGD, personality and mental illness, the references are listed as followed:
1.Torres-Rodríguez, A., Griffiths, M. D., Carbonell, X., & Oberst, U. (2018). Internet gaming disorder in adolescence: Psychological characteristics of a clinical sample. Journal of behavioral addictions, 7(3), 707-718.
2. Mallorqui-Bague, N., Fernandez-Aranda, F., Lozano-Madrid, M., Granero, R., Mestre-Bach, G., Bano, M., ... & Jimenez-Murcia, S. (2017). Internet gaming disorder and online gambling disorder: clinical and personality correlates. Journal of behavioral addictions, 6(4), 669-677.
ANSWER:
Thanks for the suggestion; we added the following information to the Introduction section:
“In that sense, several authors have analyzed the different personality and psychopathological features among IGD patients [31,32], founding less functional personality traits and higher psychopathological scores compared with a normative population.
Nevertheless, the relationship between personality and psychopathology in IGD remains unclear, and…”
- Torres-Rodríguez, A.; Griffiths, M. D.; Carbonell, X.; Oberst, U. Internet gaming disorder in adolescence: Psychological characteristics of a clinical sample. J. Behav. Addict. 2018, 7, 707–718, doi:10.1556/2006.7.2018.75.
- Mallorquí-Bagué, N.; Fernández-Aranda, F.; Lozano-Madrid, M.; Granero, R.; Mestre-Bach, G.; Baño, M.; Pino-Gutiérrez, A. Del; Gómez-Peña, M.; Aymamí, N.; Menchón, J. M.; Jiménez-Murcia, S. Internet gaming disorder and online gambling disorder: Clinical and personality correlates. J. Behav. Addict. 2017, 6, 669–677, doi:10.1556/2006.6.2017.078.
Another is the effect of aging. Actually, the maturity and development of personality during adolescence should be considered. It could be conservative to discuss the effect of personality on IGD during adolescence.
ANSWER:
Thank you for pointing out this interesting topic. We added the following information to the Discussion section:
“When interpreting the results of this study, the effects of maturity and personality development in adolescence must be taken into account. In recent research has been shown a spontaneous recovery of video-game addiction [69,70]. It must be noted that personality traits may change during individual development [71]. Therefore, it is possible that the dysfunctional personality traits found will change or disappear in the future, and with them, the influence they are having on the development or maintenance of the IGD and the associated comorbidity.”
- Przybylski, A. K.; Weinstein, N.; Murayama, K. Internet gaming disorder: Investigating the clinical relevance of a new phenomenon. Am. J. Psychiatry 2017, 174, 230–235, doi:10.1176/appi.ajp.2016.16020224.
- Rothmund, T.; Klimmt, C.; Gollwitzer, M. Low Temporal Stability of Excessive Video Game Use in German Adolescents. J. Media Psychol. 2018, 30, 53–65.
- Soto, C. J.; John, O. P.; Gosling, S. D.; Potter, J. Age Differences in Personality Traits From 10 to 65: Big Five Domains and Facets in a Large Cross-Sectional Sample. J. Pers. Soc. Psychol. 2011, 100, 330–348, doi:10.1037/a0021717.